# Clinical characteristics and laboratory features of COVID-19 in high altitude areas: A retrospective cohort study

Hanxiao Chen[1,2◦], Lang Qin[1,2◦], Sixian Wu[3], Wenming Xu[2,3], Rui Gao[1,2], Xiaohong Zhang[4]*

1 Center of Reproductive Medicine, Department of Obstetrics and Gynecology, West China Second University Hospital, Sichuan University, Chengdu, China, 2 The Key Laboratory of Birth Defects and Related Diseases of Women and Children, Ministry of Education, West China Second University Hospital, Sichuan University, Chengdu, China, 3 The Joint Laboratory for Reproductive Medicine of Sichuan University-The Chinese University of Hong Kong, West China Second University Hospital, Sichuan University, Chengdu, China, 4 Department of Emergency, Sichuan Provincial People's Hospital, Chengdu, China

◦ These authors contributed equally to this work.
* 2653099978@qq.com

**Data Availability Statement:** All relevant data are within the manuscript and its Supporting Information files.

## Abstract

Coronavirus disease 2019 (COVID-19) is highly contagious and has affected the whole world. We seek to investigate the clinical and laboratory characteristics of COVID-19 patients in the high altitude areas of Sichuan, China. In this retrospective cohort study, a total of 67 patients with laboratory-confirmed SARS-CoV-2 infections in Sichuan's Ngawa Tibetan and Qiang Autonomous Prefecture were included from February 1, 2020, to March 2, 2020. Their clinical characteristics, as well as radiological and laboratory features, were extracted. Four (6.0%) patients were categorized as severe cases; 39 (58.2%) were non-severe cases, and 24 (35.8%) were asymptomatic cases. A total of 46 (68.7%) patients were associated with cluster infection events in this study. The most common symptoms were cough, sputum production, dyspnea, fatigue or myalgia, and headache. Seven (10.4%) patients showed leucopenia, and 20 (29.9%) patients showed lymphopenia. Lymphocyte counts and neutrophil-to-lymphocyte ratios (NPR) were different between the three groups. In total, 14 (20.9%) patients had thrombocytopenia, and prothrombin times (PT) and fibrinogen levels differed between groups. We also found significant differences in sodium, chloride and calcium levels between the three groups. Antiviral therapy did not lead to obvious adverse events or shortened durations from initial positive to subsequent negative nuclei acid tests. Advanced age, hypertension, high neutrophil count, the neutrophil-to-lymphocyte ratio, fibrinogen and lactate dehydrogenase levels were identified as independent risk factors for symptomatic cases of COVID-19. In conclusion, the symptoms of patients in high altitude areas were mild, and about one third were asymptomatic. We also identified several independent risk factors for symptomatic cases of COVID-19.

**Funding:** This work was supported by the Science & Technology Department of Sichuan Province [2019JDKP0056 and 2020YFS0127] and Chengdu Science and Technology project [2020-YF05-00306-SN].

**Competing interests:** The authors have declared that no competing interests exist.

## Introduction

In December 2019, a case of pneumonia of unknown cause was detected in Wuhan, Hubei province, China [1]. The pathogen was quickly revealed as a novel betacoronavirus named the "2019 novel coronavirus" (2019-nCoV), which was subsequently renamed as the "Severe Acute Respiratory Syndrome Coronavirus 2" (SARS-Cov-2). The virus is the seventh member of the coronavirus family and belongs to the betacoronavirus 2B lineage, sharing a genome that is 96% identical with BatCov RaTG13, a SARS-like coronavirus in bats [2]. The outbreak of the disease was declared a public health emergency of international concern (PHEIC) by the World Health Organization (WHO) on January 30, 2020 [3]. Later, the WHO named it "coronavirus disease 2019" (COVID-19) on February 11, 2020 [3]. COVID-19 is highly contagious and has rapidly impacted the entire world, affecting more than 200 countries and territories.

Many studies have focused on the clinical characteristics of patients infected with SARS-CoV-2 in different countries and regions, from which we can easily see that the disease's severity and manifestation are regionally distinct [4–8]. Previous reports show reduced COVID-19 infections at high altitude [9, 10]. But it remains unknown whether there are differences in the clinical characteristics of COVID-19 patients at different altitudes. Adapted to a hypoxic environment, patients at high altitudes might also behave differently from people at lower altitudes, and treatment for patients in high altitude areas may also need special consideration [11]. There is little information, however, regarding COVID-19 in high altitude areas. Therefore, in this retrospective cohort study, we seek to provide insight into the differences in the clinical manifestation and treatment of COVID-19 between plains areas and plateau areas while also exploring the risk factors for symptomatic cases of COVID-19.

## Methods

### Data sources

We conducted a retrospective cohort study on the characteristics of laboratory-confirmed cases of COVID-19 in Sichuan's Ngawa Tibetan and Qiang Autonomous Prefecture from February 1, 2020, to March 2, 2020. All cases were diagnosed based on WHO guidelines [12] and Chinese national guidelines [13, 14]. Only the confirmed cases of SARS-CoV-2 infection—defined as positive results to high-throughput sequencing or real-time reverse-transcriptase polymerase-chain-reaction (RT-PCR) assays for nasal and pharyngeal swab specimens—were enrolled in this study. This study was reviewed and approved by the Medical Ethical Committee of Sichuan Provincial People's Hospital. The requirement of informed consent on the individual patient level was waived given the urgent need to collect clinical data and the retrospective nature of this study's design.

### Data collection

We obtained the medical records of each patient, from which we extracted epidemiological information, smoking history, comorbidities, clinical symptoms and signs, chest computed tomographic (CT) scan results and laboratory findings on admission. Epidemiological information included travel history to Wuhan within the prior 14 days, contact with confirmed cases and exposure to cluster infections. The durations from illness to first admission were also recorded. Laboratory tests consisted of a complete blood count, a coagulation test and serum biochemistry—including liver and renal function, serum proteins, creatine kinase (CK), lactate dehydrogenase (LDH), serum hypersensitive troponin I (cTnI), electrolytes, C-reactive protein (CRP), high-sensitivity C-reactive Protein (hs-CRP) and procalcitonin (PCT). Changes to

radiological findings, laboratory findings and RT-PCR assay results for each patient throughout the progression of their disease during hospitalization were also recorded in this study.

To achieve laboratory confirmation of SARS-Cov-2, pharyngeal swab specimens of each patient were collected and tested via RT-PCR assays. The SARS-Cov-2 tests for all patients were performed by a local Center for Disease Control and Prevention (CDC). Disease severity was defined based on the diagnosis and treatment of novel coronavirus pneumonia (trial version 6) by the Chinese National Health Committee. Cases were described as severe cases when they met any of the following criteria: (1) respiratory distress (a respiratory rate $\geq$ 30 breaths per minute), (2) a pulse oximeter oxygen saturation $\leq$ 93% at rest or (3) an arterial partial pressure of oxygen ($PaO_2$) divided by the fraction of inspired oxygen ($FiO_2$) $\leq$ 300 mmHg. In high-altitude areas (over 1,000 meters above sea level), this value should be corrected by the following formula: $PaO_2$ / $FiO_2$ $\times$ [Atmospheric pressure (mmHg) / 760]. Additionally, cases with chest imaging that presented significant lesion progression > 50% within 24–48 hours required management as severe cases. Specific treatments, clinical outcomes and the number of days from positive nucleic acid tests to negative nucleic acid tests for each patient were recorded. Patients were discharged from the hospital once their results from two RT-PCR tests taken one day apart were negative for the pathogen.

## Statistical analysis

Continuous variables were summarized into median and interquartile range (IQR) values, and categorical variables were expressed as frequencies and percentages. We divided the cohort into severe cases, non-severe cases and asymptomatic carriers, who were defined as asymptomatic patients with positive RT-PCR assay results and whose radiological images were normal. The Mann–Whitney U test and the Kruskal–Wallis H test were adopted to compare continuous variables between different groups, and the chi-square and Fisher's exact test were used for categorical variables as appropriate. Logistic regression analysis was employed to analyze the independent risk factors for symptomatic cases of COVID-19. Odds ratios (OR) and the corresponding 95% confidence intervals (CI) were calculated. For all analyses, a two-sided P value less than 0.05 was regarded as statistically significant. Logistic regression analysis was performed in JupyterLab, and other statistical analyses were generated using SPSS version 22.0 (IBM, Armonk, NY, USA).

## Results

### Demographics and clinical characteristics

A total of 67 patients from Sichuan's Ngawa Tibetan and Qiang Autonomous Prefecture in China (at an altitude of 2979 meters) confirmed as COVID-19 positive were included in this study, with four (6.0%) patients categorized as severe cases, 39 (58.2%) patients categorized as non-severe cases and 24 (35.8%) patients categorized as asymptomatic cases on admission. All patients were Tibetan. The median age for all patients was 40.0 years old (with an interquartile range from 20.0 to 54.7). Compared with the fact that only two (8.3%) asymptomatic patients were over 50, about half (48.7%) of the patients in the non-severe group and the majority (75%) of severe patients were older than 50 years old. Patients less than 18 years old accounted for 17.9% of the overall cases. About half (49.3%) of the patients were female. Since no patients had a direct history of exposure to Wuhan nor had contacted people from Wuhan, we assumed that all patients in this study were cases of community infection. A total of 46 (68.7%) patients were associated with cluster infection events in this study. In addition, 25 (37.3%) patients had at least one underlying disorder, the most common of which was hypertension (29.9%). Only two (3.0%) diabetes patients were identified, and three (4.5%) patients had been diagnosed

**Table 1. Demographics and clinical characteristics of patients with Coronavirus Disease 2019 (COVID-19) in high altitude areas.**

| Characteristics | All Patients (n = 67) | Disease Severity | | | |
| --- | --- | --- | --- | --- | --- |
| | | Severe Patients (n = 4) | Non-Severe Patients (n = 39) | Asymptomatic Patients (n = 24) | P Value |
| Median Age (IQR), years | 40 (20–54.7) | 60.5 (47.8–69.0) | 48.0 (37.0–63.0) | 20.0 (13.3–25.8) | < 0.001 |
| No. (%) in Age Group | | | | | |
| ≤ 18 | 12 (17.9) | 0 | 1 (2.6) | 11 (45.8) | < 0.001 |
| 19–49 | 31 (46.3) | 1 (25.0) | 19 (48.7) | 11 (45.8) | |
| 50–64 | 13 (19.4) | 1 (25.0) | 10 (25.6) | 2 (8.3) | |
| ≥ 65 | 11 (16.4) | 2 (50.0) | 9 (23.1) | 0 | |
| No. (%) by Sex | | | | | |
| Male | 34 (50.7) | 2 (50.0) | 19 (48.7) | 13 (54.2) | 0.915 |
| Female | 33 (49.3) | 2 (50.0) | 20 (51.3) | 11 (45.8) | |
| No. (%) w/ History of Exposure Within 14 Days: | | | | | |
| Cluster Infections | 46 (68.7) | 1 (25.0) | 26 (66.7) | 19 (79.2) | 0.074 |
| No. (%) by Comorbidity | | | | | |
| Any | 25 (37.3) | 3 (75.0) | 18 (46.2) | 4 (12.5) | 0.011 |
| Hypertension | 20 (29.9) | 2 (50.0) | 15 (38.5) | 3 (12.5) | |
| Diabetes | 2 (3.0) | 2 (50.0) | 0 | 0 | |
| Cardiovascular Disease | 3 (4.5) | 1 (25.0) | 2 (5.1) | 0 | |
| COPD | 2 (3.0) | 2 (50.0) | 0 | 0 | |
| Chronic Gastritis | 3 (4.5) | 0 | 2 (5.1) | 1 (4.2) | |
| Chronic Cholecystitis or Cholelithiasis | 2 (3.0) | 1 (25.0) | 0 | 1 (4.2) | |
| Other Pulmonary Disease | 3 (4.5) | 0 | 2 (5.1) | 1 (4.2) | |
| No. (%) by Signs and Symptoms | | | | | |
| Fever | 2 (3.0) | 1 (25.0) | 1 (2.6) | - | 0.434 |
| Cough | 13 (19.4) | 4 (100.0) | 9 (23.1) | - | 0.009 |
| Sputum Production | 7 (10.4) | 2 (50.0) | 5 (12.8) | - | 0.227 |
| Fatigue or Myalgia | 5 (7.5) | 1 (25.0) | 4 (10.3) | - | 0.954 |
| Headache | 5 (7.5) | 1 (25.0) | 4 (10.3) | - | 0.954 |
| Sore Throat | 1 (1.5) | 0 | 1 (2.6) | - | 1.000 |
| Dyspnea | 6 (9.0) | 3 (75.0) | 3 (7.7) | - | 0.003 |
| Median Number of Days from Illness Onset to First Admission (IQR), Days | 3.0 (1.0–7.0) | 5.0 (3.0–7.0) | 2.0 (1.0–10.0) | | 0.421 |

IQR stands for inter quartile range; COPD stands for chronic obstructive pulmonary disease. For age, sex, history of exposure and comorbidities, P values denote the comparison between severe cases, non-severe cases and asymptomatic cases. For signs and symptoms, as well as the number of days from illness onset to first admission, P values denote the comparison between severe cases and non-severe cases.

with cardiovascular disease. Two (3.0%) had chronic obstructive pulmonary disease (COPD). Three (4.5%) reported chronic gastritis. Two (3.0%) had a history of chronic cholecystitis or cholelithiasis, and three (4.5%) had other pulmonary diseases (i.e. bronchiectasis and tuberculosis). No self-reported malignant neoplasm, chronic liver disease or chronic renal disease was declared. Overall, three (75.0%) severe patients, 18 (46.2%) non-severe patients and four (12.5%) asymptomatic patients had underling comorbidities (Table 1).

The most common symptom was cough, in 13 (19.4%) cases, followed by sputum production, in seven (10.4%) cases, chest tightness or dyspnea, in six (9.0%) cases, fatigue or myalgia, in five (7.5%) cases, and headache, in five (7.5%) cases. Interestingly, only two (3.0%) patients

developed fever on admission and during hospitalization. The median number of days from the onset of the illness to first admission for all patients was three days (with an interquartile range from 1.0 to 7.0; see Table 1).

## Radiologic and laboratory findings

Of the 67 patients that had chest CT scans on admission, abnormal results were detected among all the severe cases and the majority (87.2%) of non-severe cases. The chest CT results for 24 asymptomatic patients, as described above, were normal. A total of 26 (38.8%) patients showed typical ground glass opacities, with bilateral patchy shadowing identified in nine (13.4%) cases and six (9.0%) patients showing local patchy shadowing. Interstitial abnormalities were only spotted in one (1.5%) patient (Table 2).

Seven (10.4%) patients showed leucopenia (low white blood cell counts), but we did not observe a statistical difference in white blood cell counts between severe, non-severe and asymptomatic groups (P = 0.269). A total of 20 (29.9%) patients showed lymphopenia (low lymphocyte counts), and the median lymphocyte count of the asymptomatic patients was different from those of the severe and non-severe cases (P < 0.001). Although neutrophil counts did not differ between the three groups, the neutrophil-to-lymphocyte ratio (NPR) of the asymptomatic patients was also different from those of the severe and non-severe cases (P = 0.010). Also, 14 (20.9%) patients had thrombocytopenia (low platelet counts). The prothrombin time (PT) showed statistical differences among the three subgroups (P = 0.018); four (6.0%) patients had prolonged PTs, and none of them had a PT longer than 18 seconds. Although fibrinogen levels differed between the groups (P<0.001), no patient had a fibrinogen level lower than 1 g/L. Elevated alanine aminotransferase (ALT) and aspartate aminotransferase (AST) levels were identified in 26 (38.8%) and 21 (31.3%) patients, respectively. In addition, 15 (22.4%) and 13 (19.4%) patients had decreased sodium and magnesium levels, respectively. We also witnessed significant differences in sodium, chloride and calcium levels between the three groups (P = 0.004, 0.027 and 0.003, respectively). Creatinine (Cr), CK, PCT and cTnI were normal in the majority of the cases. Elevated LDH levels were witnessed in 31 (46.3%) patients. CRP and hs-CRP levels were elevated in 11 (17.2%) and 23 (35.4%) patients, and their differences between the three groups were also statistically significant (P<0.001 and P = 0.010, respectively; see Table 2).

## Treatments and clinical outcomes

Overall, oxygen therapy, antiviral therapy, antibiotic therapy and supportive treatment were initiated in 45 (80.4%), 25 (44.6%), nine (16.1%) and 25 (44.6%) patients, respectively (Table 3). Since most of the cases were non-severe or asymptomatic cases, mechanical ventilation and systemic corticosteroids were not given. For antiviral therapy, the majority of patients received ribavirin, though some patients were switched to abidol. During the administration of antiviral therapy, we did not observe statistical variations to red blood cell (RBC) counts, WBC counts, hemoglobin (HB), platelet counts, ALT, AST or albumin (P>0.05 for all). But elevated total bilirubin (TB) levels were observed in both groups (P = 0.017 and P = 0.043, respectively). The occurrence rates of adverse effects after antiviral therapy are shown in S1 Table.

At the end of this follow-up period (February 1, 2020, to March 2, 2020), 49 (73.1%) patients were discharged from the hospital. In addition, 15 (22.4%) patients remained under hospitalization, and three (4.5%) patients, who were severe cases, were transferred to another hospital (Table 3). The median number of days from initial positive to subsequent negative

**Table 2. Radiographic and laboratory findings of patients with Coronavirus Disease 2019 (COVID-19) in high altitude areas.**

| Radiographic and Laboratory Findings | Normal Range | All Patients (n = 67) | Disease Severity | | | P Value |
|---|---|---|---|---|---|---|
| | | | Severe Patients (n = 4) | Non-Severe Patients (n = 39) | Asymptomatic Patients (n = 24) | |
| No. (%) by Chest CT Images | | | | | | |
| Abnormal | | 38 (56.7) | 4 (100.0) | 34 (87.2) | - | |
| Ground-Glass Opacity | | 26 (38.8) | 3 (75.0) | 23 (59.0) | - | |
| Local Patchy Shadowing | | 6 (9.0) | 0 | 6 (15.4) | - | |
| Bilateral Patchy Shadowing | | 9 (13.4) | 2 (50.0) | 7 (17.9) | - | |
| Interstitial Abnormalities | | 1 (1.5) | 0 | 1 (2.6) | - | |
| White Blood Cell Count ($\times10^9$/L) | 3.5–9.5 | 5.7 (4.9–6.6) | 6.8 (4.0–8.7) | 5.2 (4.8–6.4) | 6.0 (5.3–6.8) | 0.269 |
| No. / Total No. (%) $< 4\times10^9$/L | | 7/67 (10.4) | 1/4 (25.0) | 5/39 (12.8) | 1/24 (4.2) | 0.235 |
| Neutrophil Count ($\times10^9$/L) | 1.8–6.3 | 3.7 (3.0–5.0) | 5.6 (2.5–7.1) | 3.6 (3.0–5.2) | 3.9 (2.8–4.6) | 0.462 |
| Lymphocyte Count ($\times10^9$/L) | 1.1–3.2 | 1.4 (1.0–1.8) | 1.0 (0.6–1.5) | 1.2 (0.9–1.5) ** | 1.8 (1.4–2.3) | < 0.001 |
| No. / Total No. (%) $< 1.1\times10^9$/L | | 20/67 (29.9) | 3/4 (75.0) | 15/39 (38.5)** | 2/24 (8.3)*** | 0.002 |
| Neutrophil-to-Lymphocyte Ratio | | 2.6 (1.7–4.3) | 3.5 (0.2–7.1) | 3.7 (2.2–5.2) ** | 2.3 (1.4–3.1) | 0.010 |
| Hemoglobin (g/L) | 115–150 | 146.0 (130.0–159.0) | 138.5 (123.3–150.8) | 149.0 (130.0–159.0) | 147.5 (125.5–163.8) | 0.595 |
| Platelet Count ($\times10^9$/L) | 100–300 | 140.0 (105.0–197.0) | 142.5 (103.8–152.8) | 132.0 (103.0–197.0) | 149.0 (105.3–202.3) | 0.812 |
| No. / Total No. (%) $< 100\times10^9$/L | | 14/67 (20.9) | 1/4 (25.0) | 8/39 (20.5) | 5/24 (20.8) | 1.000 |
| Prothrombin Time (s) | 10–15 | 12.7 (11.9–13.4) | 12.3 (11.4–14.3) | 12.4 (11.8–13.3) ** | 13.2 (12.5–14.8) | 0.018 |
| No. / Total No. (%) >15 s | | 4 (6.0) | 0 | 0 | 4 (16.7) | 0.040 |
| Activated Partial Thromboplastin Time (s) | 22–38 | 26.9 (24.7–29.3) | 28.8 (25.3–34.2) | 27.2 (25.0–29.3) | 25.7 (23.7–29.0) | 0.462 |
| Fibrinogen (g/L) | 2–4 | 2.3 (2.0–2.9) | 3.6 (2.4–5.1) | 2.5 (2.2–2.9) ** | 1.9 (1.8–2.3) *** | < 0.001 |
| Albumin (g/L) | 35–55 | 44.0 (42.0–46.5) | 37.5 (32.3–46.8) | 44.0 (42.1–45.7) | 45.5 (41.8–47.5) | 0.241 |
| Alanine Aminotransferase (U/L) | 0–40 | 36.0 (19.0–63.0) | 16.5 (10.3–73.8) | 39.0 (24.0–63.0) | 25.0 (18.0–54.5) | 0.149 |
| No. / Total No. (%) > 40 U/L | | 26/67 (38.8) | 1/4 (25.0) | 19/39 (48.7) | 6/24 (25.0) | 0.116 |
| Aspartate Aminotransferase (U/L) | 0–40 | 32.0 (25.0–45.0) | 26.0 (17.3–75.3) | 36.0 (30.0–47.0) | 26.5 (21.5–36.8) | 0.041 |
| No. / Total No. (%) > 40 U/L | | 21/67 (31.3) | 1/4 (25.0) | 16/39 (41.0) | 4/24 (16.7) | 0.124 |
| Total Bilirubin (μmol/L) | 5.1–28 | 6.6 (4.7–13.2) | 7.8 (4.2–13.9) | 6.2 (4.6–10.3) | 10.1 (5.7–143) | 0.100 |
| Potassium (mmol/L) | 3.5–5.5 | 4.0 (3.7–4.4) | 4.1 (3.8–4.6) | 3.8 (3.6–4.4) | 4.0 (3.8–4.5) | 0.330 |
| Sodium (mmol/L) | 135–145 | 136.4 (135.3–138.0) | 130.5 (126.4–132.0) * | 136.4 (135.2–138.1) | 136.7 (136.2–137.9) *** | 0.004 |
| No. / Total No. (%) $< 135$ mmol/L | | 15/67 (22.4) | 4/4 (100.0)* | 9/39 (23.1) | 2/24 (8.3) *** | 0.001 |
| Chloride (mmol/L) | 96–108 | 104.6 (102.9–105.7) | 100.1 (92.6–102.9) * | 104.6 (103.3–105.7) | 104.6 (102.5–105.9) *** | 0.027 |
| Calcium (mmol/L) | 2.1–2.9 | 2.3 (2.2–2.3) | 2.0 (1.7–2.2) | 2.2 (2.1–2.3) ** | 2.3 (2.2–2.4) *** | 0.003 |
| Magnesium (mmol/L) | 0.75–1.02 | 0.84 (0.76–0.89) | 0.78 (0.73–0.86) | 0.84 (0.77–0.89) | 0.85 (0.75–0.90) | 0.566 |
| No. / Total No. (%) $< 0.75$ mmol/L | | 13/67 (19.4) | 2/4 (50.0) | 5/39 (7.5) | 6/24 (25.0) | 0.128 |
| Creatinine (μmol/L) | 44–123 | 60.9 (50.9–75.3) | 62.7 (52.9–82.4) | 59.4 (49.7–73.3) | 63.7 (54.2–81.8) | 0.453 |
| Creatine Kinase (U/L) | 26–174 | 64.8 (41.9–102.0) | 37.6 (26.2–79.0) | 64.4 (40.6–104.8) | 75.0 (52.7–107.3) | 0.147 |
| Lactate Dehydrogenase (U/L) | 109–245 | 247.7 (214.5–299.2) | 327.2 (173.9–485.6) | 266.6 (215.5–310.3) | 235.5 (213.5–279.9) | 0.344 |
| No. / Total No. (%) > 245 U/L | | 31/67 (46.3) | 3/4 (75.0) | 20/39 (51.3) | 8/24 (33.3) | 0.220 |
| No. / Total No. (%) C-reactive Protein (mg/L) > 10 mg/L | 0–10 | 11/66 (16.7) | 3/4 (75.0) | 8/38 (21.1) | 0*** | 0.001 |

*(Continued)*

**Table 2.** (Continued)

| Radiographic and Laboratory Findings | Normal Range | All Patients (n = 67) | Disease Severity | | | |
|---|---|---|---|---|---|---|
| | | | Severe Patients (n = 4) | Non-Severe Patients (n = 39) | Asymptomatic Patients (n = 24) | P Value |
| No. / Total No. (%) High-Sensitivity C-reactive Protein (mg/L) > 3 mg/L | 0–3 | 23/65 (35.4) | 3/4 (75.0) | 17/38 (44.7) ** | 3/23 (13.0) | 0.006 |

Data are shown as medians (with interquartile ranges) or as numbers / total numbers (and the corresponding percentage). P values denote the comparison between severe cases, non-severe cases and asymptomatic cases.

* P < 0.017 between severe and non-severe groups.

** P < 0.017 between non-severe and asymptomatic groups.

*** P < 0.017 between severe and asymptomatic groups.

nucleic acid tests was 9.0 (IQR, 3.0–11.0) days, and this figure did not differ between the antiviral therapy group and the non-antiviral therapy group (P = 0.951; see S2 Table).

## Risk factors for symptomatic cases

Several risk factors identified to be relevant to the symptomatic cases of COVID-19 were advanced age (OR, 1.076; 95% CI, 1.117–1.036), hypertension (OR, 1.013; 95% CI, 1.023–1.003), high neutrophil counts (OR, 1.462; 95% CI, 2.100–1.018), the neutrophil-to-lymphocyte ratio (OR, 1.574; 95% CI, 2.475–1.001), fibrinogen (OR, 1.927; 95% CI, 3.337–1.113) and lactate dehydrogenase levels (OR, 1.005; 95% CI, 1.010–1.001). In addition, high lymphocyte counts were related to a lower risk for symptomatic cases of COVID-19 (Table 4).

## Discussion

In this observational study, we presented the clinical and laboratory characteristics of 67 Tibetan patients confirmed with SARS-Cov-2 infections in Sichuan's Ngawa Tibetan and

**Table 3.  Treatments and outcomes of patients with Coronavirus Disease 2019 (COVID-19) in high altitude areas.**

| Treatments and Outcomes | All Patients (n = 67) | Disease Severity | | | |
|---|---|---|---|---|---|
| | | Severe Patients (n = 4) | Non-severe Patients (n = 39) | Asymptomatic Patients (n = 24) | P Value |
| No. / Total No. (%) Treatments | | | | | |
| Antiviral Therapy | 25/56 (44.6) | 2/4 (50.0) | 16/33 (48.5) | 7/19 (36.8) | 0.765 |
| Antibiotic Therapy | 9/56 (16.1) | 2/4 (50.0) | 6/33 (18.2) | 1/19 (5.3) | 0.062 |
| Oxygen Therapy | 45/56 (80.4) | 4/4 (100.0) | 26/33 (78.8) | 15/19 (78.9) | 0.885 |
| Supportive Treatment | 25/56 (44.6) | 2/4 (50.0) | 14/33 (42.4) | 9/19 (47.4) | 0.916 |
| No. (%) Outcomes: | | | | | |
| Stayed in Hospital | 15 (22.4) | 1 (25.0) | 9 (23.1) | 5 (20.8) | < 0.001 |
| Discharged from Hospital | 49 (73.1) | 0 | 30 (76.9) | 19 (79.2) | |
| Transferred to Another Hospital | 3 (4.5) | 3 (75.0) | 0 | 0 | |
| No. (%) with Two Consecutive Negative Nucleic Acid Tests for Respiratory Tract Pathogen | 49 (73.1) | 0 | 30 (76.9) | 19 (79.2) | 0.043 |
| Median Number of Days (IQR) from Initial Positive to Subsequent Negative Nucleic Acid Tests | 9.0 (3.0–11.0) | - | 9.0 (4.8–11.3) | 6.0 (3.0–10.0) | 0.216 |

Regarding treatment, P values refer to comparisons between severe cases, non-severe cases and asymptomatic cases. For nucleic acid tests, P values denote the comparison between non-severe cases and asymptomatic cases.

**Table 4. Risk factors for symptomatic cases of Coronavirus Disease 2019 (COVID-19).**

|  | Coef. | Std. Err. | OR (95% CI) | P |
|---|---|---|---|---|
| Age, Years | 0.073 | 0.019 | 1.076 (1.117–1.036) | < 0.001 |
| Hypertension | 0.013 | 0.005 | 1.013 (1.023–1.003) | 0.010 |
| White Blood Cell Count ($10^9$/L) | -0.284 | 0.098 | 0.753 (0.913–0.621) | 0.004 |
| Neutrophil Count ($10^9$/L) | 0.380 | 0.185 | 1.462 (2.100–1.018) | 0.040 |
| Lymphocyte Count ($10^9$/L) | -2.081 | 0.599 | 0.125 (0.404–0.039) | 0.001 |
| Neutrophil-to-Lymphocyte Ratio | 0.454 | 0.231 | 1.574 (2.475–1.001) | 0.049 |
| Fibrinogen (g/L) | 0.656 | 0.280 | 1.927 (3.337–1.113) | 0.019 |
| Total Bilirubin (μmol/L) | -0.163 | 0.065 | 0.849 (0.964–0.748) | 0.012 |
| Lactate Dehydrogenase (U/L) | 0.005 | 0.003 | 1.005 (1.010–1.001) | 0.033 |

Coef., coefficient; Std. Err., standard error; OR (95% CI), Odds Ratio (95% confidence interval).

Qiang Autonomous Prefecture in China. Several points must be highlighted to understand the similarities and differences in the clinical features of COVID-19 between high altitude and low altitude areas. First, the majority of cases in this study were from cluster infections, which has also been reported as a major method of transmission outside of Hubei province, China [6, 15]. Second, we found that in high altitude areas, only four (6.0%) patients were categorized as severe cases. On the other hand, 39 (58.2%) were non-severe cases, and 24 (35.8%) were asymptomatic cases with no symptoms, normal CT results but positive RT-PCR results [16]. By comparison, in the low altitude areas of China, about 26.5% of patients were diagnosed with severe COVID-19 [17]. Also, the severity of the disease was lower in high altitude areas, and our findings in this regard are consistent with another study conducted among a Peruvian population [18]. Third, we found that 75.0% of severe patients, 46.2% of non-severe patients and 12.5% of asymptomatic patients in high altitude areas had underlying comorbidities, the most common of which were hypertension and pulmonary diseases. This conformed with findings in low altitude areas [17], except for diabetes, a common comorbidity in low altitude areas that was not common in our study. Differences in diet and lifestyle may explain this result. Forth, cough, sputum production, fatigue or myalgia, headache and dyspnea were the most common symptoms in high altitude areas, but fever was not as common as in other studies conducted in low altitude areas [8, 17]. Fifth, fast changes from CT imaging were detected compared with those previously reported in Wuhan [19]. Sixth, electrolyte disturbances in these patients were also significant. Although a previous study has shown that some patients had low sodium concentrations [8], the current study revealed that the major electrolyte concentrations of sodium, chloride and calcium were altered in many cases, indicating that electrolyte disturbance is one of the important characteristics of COVID-19 patients in our cohort. In addition, the coagulation system and immune system also play essential roles in both high altitude and low altitude areas.

The coagulation function of patients in high altitude areas showed some worth-noticing changes. One noticeable result was that fibrinogen concentrations were decreased in asymptomatic cases, whereas these concentrations were significantly increased in non-severe and severe cases in our cohort. Our results were similar to those of previous studies in low altitude areas in which severe cases tended to have higher levels of fibrinogen [20]; reduced fibrinogen levels in asymptomatic cases might be correlated with high attitude. In addition, another characteristic of patients in high altitude areas was their decreased platelet levels. In our study, thrombocytopenia was more common in severe patients than in other groups, and this could be associated with SARS-CoV-2 involving the lung, where platelets are produced. Similar

results were also reported in studies on COVID-19 in low altitude areas [20]. Low platelet counts were demonstrated in severe infections, such as those with sepsis, which is related to patient prognosis. We did not, however, observe significant differences in coagulation function between non-severe patients and severe patients, as was seen in other studies [8, 15]. This was possibly due to the limited sample size of this study. Coagulation dysfunction in COVID-19 patients may be directly caused by the viral infection and indirectly stimulated by hypoxia [21], but its detailed underlying mechanism remains to be further investigated. Nevertheless, coagulation dysfunction should be carefully monitored in COVID-19 patients.

Regarding the immune system, similar to other studies performed in low altitude areas [4, 22], decreased lymphocyte counts in COVID-19 patients in the current study, especially in severe cases, demonstrate that SARS-CoV-2 might influence the immune system. In addition, we observed increased levels of NLR in severe and non-severe patients compared with asymptomatic patients. This finding agrees with the research of Qin et al., in which severe cases tended to have higher NLR levels [23]. Moreover, Liu et al. have identified that NLR is an independent risk factor for severe illness in COVID-19 patients and may be useful in the early detection of severe cases [24].

After comparing clinical and laboratory characteristics, we also investigated differences in the risk factors for COVID-19 between high altitude and low altitude areas. Advanced age, hypertension, lymphopenia, leukocytosis and elevated lactate dehydrogenase have been reported to be associated with increased odds of in-hospital death [25]. Also, advanced age, hypertension and elevated lactate dehydrogenase levels were significantly associated with the severity of COVID-19 [26]. Our results are similar to the results of previous studies. In this study, advanced age, hypertension, high neutrophil counts, a high neutrophil-to-lymphocyte ratio, high fibrinogen and high lactate dehydrogenase levels were identified as independent risk factors for symptomatic cases of COVID-19. On the other hand, high lymphocyte counts were identified as protective factors. This finding suggests that elevated neutrophil-to-lymphocyte ratios and fibrinogen may also be indicators of symptomatic cases of COVID-19.

Regarding the treatment of COVID-19 patients, the majority of patients received oxygen therapy and antiviral therapy, which was consistent with studies in low altitude areas [17]. The present study did not report obvious adverse drug-related events among patients receiving antiviral therapy, including the use of ribavirin and abidol. The median number of days from initial positive to subsequent negative nuclei acid tests was 9.0 (IQR, 3.0–11.0) days, and this may have been associated with the recovery of the lymphocyte and immune system. But this duration did not differ between the antiviral treatment group and the non-antiviral treatment group. The question of whether the clinical application of antiviral drugs can improve the prognosis of patients remains uncertain and warrants further exploration [6]. At present, current clinical studies have revealed that severe and critical COVID-19 may cause poor gas exchange in the lungs, leading to hypoxia [27]. Therefore, oxygen therapy plays an important role in the treatment of the disease. We thus summarize the main differences between high altitude areas and low altitude areas as follows: (1) Patients with SARS-CoV-2 infection have lower oxygen saturation in plateau areas, and hypoxia may be more obvious. (2) COVID-19 patients have a higher probability of progressing to pulmonary edema and brain edema in plateau areas. Furthermore, several points need to be addressed regarding the use of oxygen therapy when treating COVID-19 in high altitude areas. First, attention should be paid to oxygen concentrations and the method of oxygen delivery. Low flow oxygen therapy is recommended; that is, oxygen flow rates should be maintained at 2–4 L per minute to ensure the efficacy of oxygen therapy and also to avoid side effects. Maintaining a continuous pattern of oxygen delivery, instead of an intermittent pattern, is also important. Regarding the characteristics of the disease, oxygen therapy should be scheduled in the afternoon or the evening. Second,

physicians must be alert to the occurrence of cerebral edema, especially in children, and they must pay attention to the amount and speed of intravenous fluid therapy to prevent cerebral edema. Last but not least, in high-altitude areas (over 1,000 meters above sea level), as described above, the $PaO_2/FiO_2$ value should be corrected. The implementation of these measures in oxygen therapy will increase the efficacy of COVID-19 treatment in plateau areas. Also, the interaction between hypoxia and the virus is complicated, and the underlying mechanisms may be different for different viruses. Some studies have reported that hypoxia may inhibit or stimulate the propagation of viruses and that HIF-1α is essential in modulating this process [28]. In this regard, more studies regarding hypoxia are needed to unearth the relevant mechanisms and provide a potential treatment target for COVID-19.

Our study has several limitations, and the findings should be interpreted with caution. First, only a small sample of 67 patients was included in the current study, and the number of severe cases was also limited. Thus, a lack of statistical significance might not rule out differences between the subgroups. Second, the patients in this study were all from one hospital in Sichuan province, where medical supplies and equipment may have differed from those of other regions. Also, we were not able to compare the clinical characteristics of patients in high altitude areas with those of patients in low altitude areas due to an inability to access the relevant data. Nonetheless, we tried our best to precisely describe all the clinical and laboratory features of COVID-19 in high altitude areas, allowing an adequate comparison with other previously published studies conducted in low altitude areas. Third, since data on D-dimer results and specific subsets of lymphocytes were not available, further studies are required to investigate the role of coagulation and immune response in COVID-19 patients.

In conclusion, this study presented a detailed description of the clinical and laboratory characteristics of COVID-19 patients in high altitude areas, thus helping clinicians better manage patients.

## Supporting information

**S1 Table. The adverse effects of antiviral drugs.**
(DOCX)

**S2 Table. Patient outcomes between the antiviral therapy group and the non-antiviral therapy group.**
(DOCX)

**S1 Dataset.**
(XLSX)

## Acknowledgments

We would like to thank all patients included in this study. We thank Lan Ma and Zheng Wang from Sichuan Orient Software Technology Company Limited for statistical guidance.

## Author Contributions

**Conceptualization:** Xiaohong Zhang.

**Data curation:** Xiaohong Zhang.

**Formal analysis:** Sixian Wu.

**Methodology:** Xiaohong Zhang.

**Project administration:** Lang Qin.

**Supervision:** Lang Qin.

**Validation:** Rui Gao.

**Writing – original draft:** Hanxiao Chen.

**Writing – review & editing:** Wenming Xu.

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
