## [Decision Letter · Decision Letter 0]

23 Feb 2021

PONE-D-20-33764

Clinical characteristics and laboratory features of COVID-19 in high altitude areas: A retrospective cohort study

PLOS ONE

Dear Dr. Chen,

Thank you for submitting your manuscript to PLOS ONE. After careful consideration, we feel that it has merit but does not fully meet PLOS ONE’s publication criteria as it currently stands. Therefore, we invite you to submit a revised version of the manuscript that addresses the points raised during the review process.

The reviewers have commented on your above paper. They have suggested that this manuscript be revised according to the reviewers suggestions and resubmitted.  Provided you address the changes recommended, the manuscript will be accepted for publication.

We look forward to receiving your revised manuscript.

Kind regards,

Prof. Raffaele Serra, M.D., Ph.D

Academic Editor

PLOS ONE

Journal Requirements:

Additional Editor Comments:

The reviewers have commented on your above paper. They have suggested that this manuscript be revised according to the reviewers suggestions and resubmitted.

Reviewers' comments:

Reviewer's Responses to Questions

**Comments to the Author**

1. Is the manuscript technically sound, and do the data support the conclusions?

Reviewer #1: Yes

2. Has the statistical analysis been performed appropriately and rigorously? 

Reviewer #1: Yes

3. Have the authors made all data underlying the findings in their manuscript fully available?

Reviewer #1: Yes

4. Is the manuscript presented in an intelligible fashion and written in standard English?

Reviewer #1: No

5. Review Comments to the Author

Reviewer #1: The authors describe the clinical and laboratory characteristics of COVID-19 patients in high altitude areas of Sichuan province, China. According to the results present in this manuscript, the authors do not compare the patients in high and low altitude areas and only described a cohort study. Without the information of patients positive to COVID-19 in-plane area, it is difficult to correlate the symptoms and attribute some aspects with the geographic area. The results are clear and will be contributed to the knowledge of COVID-19 disease.

The manuscript must be reviewed at the grammatical and typographical levels (example: a lack of spaces between the words)

Although some aspects of these results must be improved, I recommend that this manuscript be revised.

6. PLOS authors have the option to publish the peer review history of their article (what does this mean?). If published, this will include your full peer review and any attached files.

Reviewer #1: No

---

## [Author Response · Author response to Decision Letter 0]

23 Mar 2021

For Journal Requirements:

Response: We confirm that our manuscript meets PLOS ONE's style requirements.

Response: There are no restrictions of the data. We have uploaded the data as Supporting Information files.

Response: We added the following references according to revision:

11. Breevoort A, Carosso GA, Mostajo-Radji MA. High-altitude populations need special considerations for COVID-19. Nat Commun. 2020;11(1):3280. Epub 2020/07/03. doi: 10.1038/s41467-020-17131-6. PubMed PMID: 32612128.

17. Fu L, Wang B, Yuan T, Chen X, Ao Y, Fitzpatrick T, et al. Clinical characteristics of coronavirus disease 2019 (COVID-19) in China: A systematic review and meta-analysis. J Infect. 2020;80(6):656-65. Epub 2020/04/14. doi: 10.1016/j.jinf.2020.03.041. PubMed PMID: 32283155; PubMed Central PMCID: PMCPMC7151416.

18. Seclén SN, Nunez-Robles E, Yovera-Aldana M, Arias-Chumpitaz A. Incidence of COVID-19 infection and prevalence of diabetes, obesity and hypertension according to altitude in Peruvian population. Diabetes Res Clin Pract. 2020;169:108463. Epub 2020/09/25. doi: 10.1016/j.diabres.2020.108463. PubMed PMID: 32971150; PubMed Central PMCID: PMCPMC7505740.

20. Qian GQ, Yang NB, Ding F, Ma AHY, Wang ZY, Shen YF, et al. Epidemiologic and clinical characteristics of 91 hospitalized patients with COVID-19 in Zhejiang, China: a retrospective, multi-centre case series. Qjm. 2020;113(7):474-81. Epub 2020/03/18. doi: 10.1093/qjmed/hcaa089. PubMed PMID: 32181807; PubMed Central PMCID: PMCPMC7184349.

We also exclude the following references according to revision:

22. Wu PF, Li RZ, Zhang W, Hu HY, Wang W, Lin Y. Polycystic ovary syndrome is causally associated with estrogen receptor-positive instead of estrogen receptor-negative breast cancer: a Mendelian randomization study. American journal of obstetrics and gynecology. May 13 2020;doi:10.1016/j.ajog.2020.05.016

We updated the following references which were from medRxiv or bioRxiv:

24. Liu J, Liu Y, Xiang P, Pu L, Xiong H, Li C, et al. Neutrophil-to-lymphocyte ratio predicts critical illness patients with 2019 coronavirus disease in the early stage. J Transl Med. 2020;18(1):206. Epub 2020/05/22. doi: 10.1186/s12967-020-02374-0. PubMed PMID: 32434518; PubMed Central PMCID: PMCPMC7237880.

26. Huang H, Cai S, Li Y, Li Y, Fan Y, Li L, et al. Prognostic Factors for COVID-19 Pneumonia Progression to Severe Symptoms Based on Earlier Clinical Features: A Retrospective Analysis. Front Med (Lausanne). 2020;7:557453. Epub 2020/10/31. doi: 10.3389/fmed.2020.557453. PubMed PMID: 33123541; PubMed Central PMCID: PMCPMC7571455.

Reviewers' Comments to the Author:

Reviewer #1: The authors describe the clinical and laboratory characteristics of COVID-19 patients in high altitude areas of Sichuan province, China. According to the results present in this manuscript, the authors do not compare the patients in high and low altitude areas and only described a cohort study. Without the information of patients positive to COVID-19 in-plane area, it is difficult to correlate the symptoms and attribute some aspects with the geographic area. The results are clear and will be contributed to the knowledge of COVID-19 disease. The manuscript must be reviewed at the grammatical and typographical levels (example: a lack of spaces between the words)

Although some aspects of these results must be improved, I recommend that this manuscript be revised.

Response: First of all, thank you for bring up this important issue. We are well aware of the limitations of our study which were that we were not able to compare the characteristics of COVID-19 patients in high and low altitude areas due to the inability to access the relevant data. Nonetheless, we tried our best to precisely describe all the clinical and laboratory features of COVID-19 in high altitude areas, allowing an adequate comparison with other previously published studies conducted in low altitude areas. Also, we described the similarities and differences in the clinical features of COVID-19 between high and low altitude areas in discussion part of the manuscript in detail.

Moreover, thanks for your suggestion of language editing of the manuscript. We have tried our best to polish the language in the revised manuscript. We believe the language has been improved in the updated version.

---

## [Editor Report · Decision Letter 1]

29 Mar 2021

Clinical characteristics and laboratory features of COVID-19 in high altitude areas: A retrospective cohort study

PONE-D-20-33764R1

Dear Dr. Zhang,

We’re pleased to inform you that your manuscript has been judged scientifically suitable for publication and will be formally accepted for publication once it meets all outstanding technical requirements.

Kind regards,

Prof. Raffaele Serra, M.D., Ph.D

Academic Editor

PLOS ONE

Additional Editor Comments (optional):

amended manuscript is acceptable
---

## [Editor Report · Acceptance letter]

10 May 2021

PONE-D-20-33764R1 

Clinical characteristics and laboratory features of COVID-19 in high altitude areas: A retrospective cohort study 

Dear Dr. Zhang:

I'm pleased to inform you that your manuscript has been deemed suitable for publication in PLOS ONE. Congratulations! Your manuscript is now with our production department. 

Kind regards, 

on behalf of

Prof. Raffaele Serra 

Academic Editor

PLOS ONE